# Roles for Autophagy in Esophageal Carcinogenesis: Implications for Improving Patient Outcomes

**DOI:** 10.3390/cancers11111697

**Published:** 2019-10-31

**Authors:** Reshu Saxena, Alena Klochkova, Mary Grace Murray, Mohammad Faujul Kabir, Safiyah Samad, Tyler Beccari, Julie Gang, Kishan Patel, Kathryn E. Hamilton, Kelly A. Whelan

**Affiliations:** 1Fels Institute for Cancer Research & Molecular Biology, Lewis Katz School of Medicine at Temple University, Philadelphia, PA 19140, USA; tuj81303@temple.edu (R.S.); tuj51051@temple.edu (A.K.); mary.murray337@gmail.com (M.G.M.); tul69249@temple.edu (M.F.K.); ssamad1999@gmail.com (S.S.); tbeccari@holyfamily.edu (T.B.); tug67451@temple.edu (J.G.); tug99040@temple.edu (K.P.); 2Department of Pediatrics, Division of Gastroenterology, Hepatology, and Nutrition, Children’s Hospital of Philadelphia, University of Pennsylvania Perelman School of Medicine, Philadelphia, PA 19104, USA; HAMILTONK1@email.chop.edu; 3Department of Pathology & Laboratory Medicine, Lewis Katz School of Medicine at Temple University, Philadelphia, PA 19140, USA

**Keywords:** esophageal cancer, autophagy, esophagus, esophageal squamous cell carcinoma, esophageal adenocarcinoma

## Abstract

Esophageal cancer is among the most aggressive forms of human malignancy with five-year survival rates of <20%. Autophagy is an evolutionarily conserved catabolic process that degrades and recycles damaged organelles and misfolded proteins to maintain cellular homeostasis. While alterations in autophagy have been associated with carcinogenesis across tissues, cell type- and context-dependent roles for autophagy have been reported. Herein, we review the current knowledge related to autophagy in esophageal squamous cell carcinoma (ESCC) and esophageal adenocarcinoma (EAC), the two most common subtypes of esophageal malignancy. We explore roles for autophagy in the development and progression of ESCC and EAC. We then continue to discuss molecular markers of autophagy as they relate to esophageal patient outcomes. Finally, we summarize current literature examining roles for autophagy in ESCC and EAC response to therapy and discuss considerations for the potential use of autophagy inhibitors as experimental therapeutics that may improve patient outcomes in esophageal cancer.

## 1. Introduction

Esophageal cancer is the eighth most prevalent cancer type and the sixth-leading cause of cancer-associated mortality worldwide [1]. Esophageal adenocarcinoma (EAC) and esophageal squamous cell carcinoma (ESCC) comprise the two primary histological subtypes of esophageal malignancy. ESCC arises via malignant transformation of esophageal epithelial cells with activation of epidermal growth factor receptor (EGFR) and cyclin D1 oncogenes and mutations in the tumor suppressor gene *TP53* representing common genetic alterations [2,3,4,5,6]. By contrast, EAC develops as esophageal epithelium is displaced by specialized intestinal columnar mucosa. This metaplastic condition, termed Barret’s esophagus (BE), arises in the setting of gastroesophageal reflux disease (GERD). BE may progress to dysplasia which further enhances EAC risk [7]. Geographic distribution has been noted for esophageal cancer incidence. ESCC occurs most frequently in Africa and East Asia while EAC rates have dramatically increased in Western nations, including the United States, in recent decades. Despite marked differences in epidemiology and pathophysiology, both ESCC and EAC display five-year survival rates of <20% that are associated with late stage diagnosis, frequent metastasis and therapeutic resistance [8,9,10]. As such, there exists an urgent need for the development of novel approaches for esophageal cancer therapy.

Autophagy is a highly conserved catabolic process through which cellular constituents are sequestered by autophagic vesicles (AVs) then delivered to lysosomes for hydrolytic degradation. The molecular regulation of autophagy is complex as detailed in Figure 1. Mammalian target of rapamycin complex 1 (mTORC1) and AMP-activated protein kinase (AMPK) are two well-established regulators of autophagy that act to modulate Unc-51-like autophagy activating kinase 1 (ULK1)-mediated nucleation of AVs. Elongation and maturation of AVs are subsequently mediated by various autophagy-related (ATG) proteins. Following fusion with lysosomes, autophagic cargo is broken down, providing substrates for macromolecule biosynthesis.

Autophagy occurs at a basal level in most tissue types, including the esophagus [11,12], and has been shown to be induced in response to a variety of stressors, including starvation, hypoxia, and inflammation. Autophagy has been implicated in a variety of human diseases with context- and tissue-dependent roles. With regard to cancer, the role of autophagy is complex. Autophagy serves a tumor suppressor role early in carcinogenesis. In established tumors, however, autophagy acts as a tumor-promoting factor that aids survival in the harsh tumor microenvironment as well as in response to therapy-associated stress [13]. The reliance of tumor cells on autophagy for survival has been discussed as a potential Achilles’ heel that may be leveraged to kill tumor cells either as monotherapy or in conjunction with current stand of care protocols. While studies utilizing the lysosomotropic autophagy inhibitor hydroxychloroquine (HCQ) have shown varying levels of success in recent clinical trials across cancer types [14], there are presently no clinical trials examining the impact of autophagy modulation in esophageal cancer.

Herein, we aim to review the current literature related to autophagy and esophageal cancer, both ESCC and EAC. As our understanding of the functional role of autophagy in esophageal biology under conditions of health and disease continues to emerge, this information may aid in the design of autophagy-targeting therapeutic strategies with potential to improve outcomes for esophageal cancer patients.

## 2. Roles for Autophagy in Esophageal Carcinogenesis

Although ESCC and EAC are commonly classified as ‘esophageal cancer’, distinctions in epidemiology and pathophysiology exist between these disease states, and emerging genetic studies indicate that ESCC lesions more closely resemble other squamous cell carcinomas than EAC lesions [4,15]. Given these findings and the context- and cell type-dependent nature of autophagy, it is important to investigate the specific roles that autophagy may play in development, progression, and therapeutic response of ESCC and EAC individually.

Studies utilizing experimental model systems have implicated autophagy in both pro- and anti-tumor responses activated in response to various cellular and microenvironmental factors associated with esophageal cancer (Figure 2). Amongst cellular factors that have been shown to induce AVs in ESCC there is a common convergence on ULK1, a serine/threonine kinase essential for AV formation [16,17], via modulation of several upstream signaling pathways. AMPK is a well-established activator of autophagy via direct phosphorylation of ULK1, as well as phosphorylation of ULK-interacting proteins [18]. The AMPK/ULK1 axis has been implicated in activation of autophagy induced by the tumor metastasis-associated factor, metastasis-associated colon cancer-1 (MACC1). In vitro studies using the ESCC cell lines Eca9706 and KYSE150, demonstrated that MACC1 facilitates ESCC proliferation, migration, and invasion in an autophagy-dependent manner [19]. In this context, autophagy activation was determined to be mediated by AMPK as the AMPK activator Metformin rescued ULK1 phosphorylation. Autophagy activation downstream of the tight junction protein Claudin 1 was also recently demonstrated to involve AMPK/ULK1. In the ESCC cell lines TE10 and TE11, overexpression of the tight junction protein Claudin 1 induced autophagy concurrent with activation of AMPK. In this system, downregulation of signal transducer and activator of transcription (STAT) 1, an inhibitor of ULK1 transcription [20], was also found. Autophagy was implicated in Claudin 1-mediated increases in proliferation, migration, and invasion in these ESCC cell lines as these phenotypes were abrogated by 3-methyladenine (3MA), a class III PI3 Kinase (PI3K) inhibitor that blocks AV formation. Interestingly, while Claudin 1 overexpression also promoted AMPK activation in primary normal esophageal epithelial cells, downstream autophagy induction was not observed. Thus, Claudin 1 may have different functions in normal cells as compared to their malignant counterparts. In these two studies, the authors further demonstrated that MACC1 or Claudin 1 expression correlated with lymph node metastasis in ESCC patients; however, the relevance of the AMPK/ULK1/autophagy signaling to these in vivo observations remains to be determined.

While AMPK activates ULK1, mTORC1 limits ULK1 activation [18]. A recent study linked the histone deacetylase SIRT6 to activation of autophagy flux in via modulation of the mTORC1/ULK1 axis [21]. The authors demonstrated that SIRT6 overexpression in the ESCC cell lines Ec109 and TE1 increased cell proliferation concurrent with diminished mTORC1 phosphorylation and augmented ULK1 phosphorylation. Binding of SIRT6 to ULK1 was also observed; however, the functional consequences of this binding with regard to autophagy flux has yet to be determined. Additionally, it is unclear whether SIRT6-mediated suppression of mTORC1 is dependent upon the protein’s histone deacetylase activity. Although SIRT6-mediated histone deacetylation has been demonstrated to be indispensable for attenuation of mTORC1 signaling downstream of insulin growth factor receptor-1 in bronchial epithelial cells responding to cigarette smoke [22], a recent study showed that SIRT6 may inactivate mTORC1 indirectly via binding to the transcription factor Sp1 [23]. Thus, additional investigation into SIRT6-mediated regulation of the mTORC1/ULK1/autophagy axis is required in ESCC. MicroRNA (miR)-503-mediated proliferation, migration, and invasion of Eca109 and Eca9706 ESCC cells has also been linked to autophagy activation occurring in response to diminished mTORC1 activation [24]. In this study, miR-305 was shown to activate autophagy flux through direct binding to a novel target transcript, protein kinase CAMP-activated catalytic subunit alpha (PRKACA). The resultant downregulation of PRKACA was associated with decreased phosphorylation of mTORC1 and increased autophagy flux. PRKACA is a known activator of protein kinase A (PKA) which in turn activates mTORC1. As such, the authors conclude that miR-503 induces autophagy flux in ESCC via miR-503-mediated inactivation of PKA/mTORC1 signaling.

Microenvironmental factors have also been linked to autophagy activation in ESCC. Bcl-2/adenovirus E1B 19-kDa interacting protein (BNIP3)-mediated autophagy suppresses cell death in the ESCC cell lines CAES17 and KYSE40 [25]. Additionally, we have reported that autophagy is activated in transformed EPC2T esophageal keratinocytes upon exposure to transforming growth factor β (TGFβ) [26], a cytokine produced by both ESCC cells and the surrounding stroma. In this context, autophagy clears damaged mitochondria to regulate redox balance during TGFβ-mediated epithelial–mesenchymal transition (EMT). This promotes generation of CD44^high^ cancer stem cells (CSCs) with increased malignant potential. A recent study has linked ATG7 to stemness in human bladder cancer cells via stabilization of CD44 protein [27]. Although this mechanism has not been explored in ESCC or EAC, in which CD44 is also a CSC marker [28], it suggests that ATG proteins may modulate stemness independent of their functions in autophagy. An additional study linked autophagy flux to maintenance of OV6+ CSCs in ESCC [29]; however, the signals promoting ATG7-dependent autophagy in these cells were not explored. Rather, the authors demonstrated a role for autophagy in Wnt/β-catenin signaling in OV6+ CSCs using Eca109 and TE1 ESCC cell lines. Thus, autophagy may be regulated by signals inducing CSC generation and maintenance while also contributing to the regulation of these signals.

While the aforementioned studies describe activation of autophagy as a mechanism to promote ESCC carcinogenesis, inactivation of autophagy by the DNA repair protein RAD51 has been reported to enhance proliferation in ESCC [30]. In this study, RAD51 was shown to be highly expressed in human ESCC tumors and to promote xenograft tumor growth of Ec109 ESCC cells. In vitro studies further demonstrated that the oncogenic effects of RAD51 require the checkpoint kinase CHK1, which is subjected to degradation by chaperone-mediated autophagy [31]. RAD51 loss increased autophagy flux and 3MA-mediated autophagy inhibition restored CHK1 expression and soft agar colony formation in RAD51-depleted Ec109 and Ec9706 ESCC cells, respectively. The authors continued to demonstrate that Ec109 cells expressing RAD51 exhibit cell death upon treatment with the autophagy inhibitors 3MA and HCQ while their RAD51-deficient counterparts fail to respond. These studies present autophagy inhibitors as a potential approach to ESCC therapy in patients with RAD51 upregulation; however, additional preclinical studies are necessary to further investigate the efficacy of such an approach.

Although the role of autophagy in GERD and its sequelae has not been as well-studied, alterations in autophagy have been identified in esophageal epithelium upon exposure to GERD-associated stimuli. Indeed, we have reported that acid exposure in normal immortalized esophageal keratinocytes increases AV content as a mechanism to limit oxidative stress and cell death [11]. Basal AV content was augmented in the CPA cell line, which is derived from nondysplastic BE lesions. Furthermore, CPA cells displayed resistance to pH stress-induced oxidative stress. Interestingly, resistance to pH-mediated cell stress was maintained in cell lines derived from either dysplastic BE or EAC in an autophagy-dependent manner. Thus, while autophagy may be critical for adaptation to the GERD microenvironment, the process may be dispensable later in the progression to EAC. In line with this notion, Roesly and colleagues demonstrated that acute deoxycholic acid treatment increased expression of Beclin-1, a critical mediator of AV formation and maturation, and AV content in BE-derived cells. This effect was lost upon chronic deoxycholic acid exposure [32]. Furthermore, evidence of increased autophagy was detected in endoscopic biopsies from human BE patients as compared to both normal subjects and those with established EAC. Paligenosis is an injury-associated reprogramming of differentiated cells involving sequential upregulation of autophagy, metaplastic gene expression and mTORC signaling [33]. It has recently been proposed that paligenosis may contribute to esophageal metaplasia and EAC pathogenesis by allowing for accumulation of mutations under conditions of chronic reflux [34]. It will be of interest to examine paligenosis as a mediator of BE and EAC and also to determine what role autophagy may play in this context.

The described studies clearly implicate autophagy in the pathogenesis of both ESCC and EAC. Presently, our understanding of the detailed molecular mechanisms through which autophagy is impacted by stimuli relevant to these two subtypes of esophageal cancer remains limited. Additionally, the consequences of these alterations in autophagy as they relate to esophageal carcinogenesis has yet to be fully elucidated.

In line with the described studies demonstrating autophagy alterations in model systems of esophageal cancer, various studies have shown changes in expression of autophagy markers in ESCC and EAC patients. Several of these studies have identified significant associations with patient survival (Table 1). Microtubule-associated light chain 3 (LC3), a protein that undergoes posttranslational cleavage and lipidation prior to incorporation into AVs, is the most well-characterized autophagy marker in esophageal cancer. In ESCC, at least seven independent studies have examined LC3 levels in relation to patient survival with three demonstrating that increased LC3 is associated with poor overall survival [26,35,36], while two studies found the opposite- increased LC3 corelates with better overall survival [37,38]. The remaining two studies failed to identify any correlation between LC3 expression and survival [39,40]. Although each of these seven studies utilized immunohistochemistry to detect LC3, there was variability in the specific antigen targeted with six studies evaluating full length LC3, LC3A, or LC3B [35,36,37,39,40], while our own study evaluated expression of cleaved LC3A [26]. Examination of intracellular LC3 staining patterns may help to further clarify the protein’s role as a biomarker in esophageal cancer. Indeed, the relationship between LC3 and EAC patient outcome varied among three identified staining categories—diffuse cytoplasmic, crescent or ring-like structures, and globular structures—in a study by El-Mashed and colleagues [41]. Specifically, low diffuse cytoplasmic LC3 staining or high staining for either of the other two LC3 staining patterns was associated with poor overall survival. An additional approach to improving upon the efficacy of LC3 as a biomarker in esophageal cancer is to evaluate its expression in combination with additional autophagy-associated proteins. In one study, neither expression of LC3B nor the tumor suppressor protein p53 alone correlated with ESCC patient outcome; however, patients with high co-expression of LC3 and p53 displayed decreased five-year survival as compared to those with low co-expression [39]. Expression of LC3 alone failed to reach statistical significance in terms of association with EAC survival; however, patients with low levels of both LC3B and p62, an autophagy cargo-identifying protein, fared worse than those with high co-expression or ‘mixed’ type lesions (displaying high expression of LC3 and low expression of p62 or vice versa) [42].

Beyond LC3, the autophagy-associated proteins p62, Beclin-1 and ULK1 have been studied in human esophageal cancer patients. p62 expression was found to be elevated in EAC lesions as compared to normal esophageal mucosa; however, patients with tumors expressing low levels of p62 exhibited decreased overall survival [42], which may be reflective of altered functions for autophagy throughout carcinogenesis. As is the case with LC3, consideration of intercellular patterns of p62 staining may reveal additional associations with clinicopathological information. When compared to primary EAC lesions, p62 dot-like staining was found to be elevated in distant metastases whereas lymph node metastases exhibited higher nuclear p62 levels [42]. By contrast, no difference was seen in cytoplasmic p62 staining when comparing primary tumors to either lymph node or distant metastases. With regard to patient survival, low levels of both cytoplasmic and dot-like staining for p62, but not nuclear staining, were significantly associated with worse outcome [42]. In both ESCC and EAC, Beclin-1 expression is decreased in tumors as compared to normal mucosa [32,43,44]. Although the association between Beclin-1 expression and survival in EAC remains to be determined, three independent studies have evaluated this relationship in ESCC. Two of these studies found that negative Beclin-1 expression is associated with poor overall survival [43,44]. The third study failed to define any significant association between Beclin-1 alone and clinical outcome in ESCC, but demonstrated that negative co-expression of Beclin-1 and LC3 correlated with poor survival as compared to positive co-expression of the two proteins [35]. Similarly, ULK1 expression was found to be higher in ESCC tissues as compared to non-cancerous esophageal tissue in two independent studies. One study demonstrated that patients with low ULK1 had decreased survival rates [45], while the other showed that patients with high expression of ULK1 had lower survival rates [46].

In addition to immunostaining staining of autophagy markers in esophageal cancer patient tissues samples, genomic studies have identified alterations in genes associated with the autophagy pathway in both ESCC and EAC. A large scale integrated genomic characterization of esophageal cancer revealed three subtypes, ESCC1-3, two of which displayed deep deletions in *ATG7,* the protein product of which mediates LC3 lipidation and AV formation [4]. In ESCC1, which was most commonly identified in Asian patients and characterized by mutations in the NRF2 oxidative stress response pathway, *ATG7* deletions were found in 12% of patients. Frequency of *ATG7* deletions was 25% in ESCC3, which was comprised exclusively of North American patients and displayed genetic alterations that are predicted to activate PI3K signaling. *ATG7* deletions were absent in the ESCC2 subtype, which was identified most frequently in Eastern European and South American patients and in which *NOTCH1* mutations were most prevalent. *Atg7* in mice is essential for autophagy-mediated survival during weaning; however, autophagy activation is detected in the absence of Atg7 in these animals [46,47]. Additional studies in ESCC subtypes 1–3 are necessary to determine how ATG7 loss impacts autophagy levels. In EAC, a fusion event has been identified between the genes encoding vacuole membrane protein 1 (VMP1), a protein that promotes AV formation via binding to Beclin-1, and ribosomal protein S6 kinase (RPS6KB1), a downstream effector of mTORC1 signaling [48]. Two protein products resulting from this *RPS6KB1-VMP1* fusion were identified in ~10% of EAC tumors, with each lacking the full kinase domain of RPS6KB1 and containing a truncated form of VMP1. Functional analysis of one of the RPS6KB1-VMP1 fusion proteins revealed inhibition of VMP1 binding to Beclin-1 and decreased levels of autophagy under basal conditions and in response to nutrient deprivation. Expression of RPS6KB1-VMP1 in CPA cells, which were derived from non-dysplastic BE, resulted in increased cell growth, consistent with autophagy acting as a tumor suppressive factor early in BE. Although the impact of this fusion protein upon dysplastic BE and EAC has not yet been explored, RPS6KB1-VMP1-positive EAC patients displayed poorer survival as compared to not expressing the fusion protein.

While evaluation of autophagy-associated genes and proteins in human patients has potential to be informative with regard to esophageal cancer patient outcomes, additional work needs to be performed to determine how to best integrate evaluation of autophagy into clinical practice. From the described immunohistochemistry-based studies, it is clear that selection of the autophagy marker or markers to be studied must be considered carefully as associations with clinical data vary among these proteins. However, ambiguity also exists when considering a single autophagy marker as a biomarker, as is evident in the described studies related to LC3, Beclin-1, and ULK-1 in ESCC. Given the presence of deep deletions in the *ATG7* gene [47] and the presence of *RPS6KB1-VMP1* gene fusions [48] in subsets of esophageal cancer patients, it may be critical to take into account the impact of patient genetics upon the use of autophagy factors as biomarkers in esophageal cancer.

## 3. Roles for Autophagy in Esophageal Cancer Responses to Therapy

Activation of autophagy has been implicated as mechanism of resistance to various therapies across tumor types. Given the described studies implicating autophagy in esophageal carcinogenesis, it is of interest to investigate the role of autophagy in therapy response in ESCC and EAC. This is particularly pertinent in light of the refractory nature of esophageal cancer to current standard of care, which includes surgery often in conjunction with chemo- and radiotherapy. As our understanding of the role of autophagy in esophageal cancer response to therapy emerges, autophagy-targeted approaches to improving patient outcomes in ESCC and EAC may be implemented.

The commonly used chemotherapeutic agents cisplatin and 5-fluouracil (5FU) induce autophagy in both ESCC and EAC cell lines in vitro [49,50,51,52,53,54]. In one study, the ESCC cell line OE21 and the EAC cell line OE33 were demonstrated to exhibit sensitivity to both 5FU and cisplatin that was associated with apoptosis and a lack of autophagy induction [53]. By contrast, autophagy was induced in the ESCC cell line KYSE450 and the EAC cell line OE19, both of which exhibited resistance to 5FU and cisplatin. In KYSE450, genetic inhibition of Beclin-1 and ATG7 together decreased AV induction following 5FU treatment concurrent with diminished cell viability, indicating that autophagy contributes to chemoresistance. Interestingly, this study further revealed that neither 3MA nor the lysosomotropic agents CQ and Bafilomycin A1 impacted 5FU sensitivity in KYSE450 cells. Thus, pharmacological agents that indirectly target the autophagy pathway may not always mimic the effects of genetic impairment of the pathway. In a follow-up to this study, apoptosis in the 5FU-sensitive OE21 and OE33 cell lines was found to be associated with upregulation of the ubiquitin-like protein modifier interferon-stimulated gene 15 (IGS15), which acted to block autophagy [50]. The mechanisms through which IGS15 regulates autophagy as well as the therapeutic applications of targeting IGS15 in the context of 5FU treatment remain to be determined. In the ESCC cell line Ec109, autophagy in response to cisplatin was concurrent with diminished activation of mTORC1. In Ec109 xenograft tumors, cisplatin in combination with CQ-mediated autophagy inhibtion decreased growth compared to vehicle whereas neither agent alone had a significant impact, indicating a protective effect of autophagy in response to cisplatin [55].

Using a 3D organoid platform, we have recently identified induction of CD44^high^ CSCs with enhanced autophagy flux in 5FU-resistant TE11 ESCC cells [49]. These findings complement our previous work demonstrating that autophagy is a permissive factor in EMT-mediated CSC expansion in the ESCC tumor microenvironment [26]. In this study, we reported that the lysosomotropic drugs CQ and Lys05 decreased CD44^high^ CSCs in TE11 ESCC xenograft tumors concurrent with increased oxidative stress. Although ongoing studies aim to determine the impact of autophagy inhibition upon ESCC response to therapy, we speculate that administration of autophagy inhibitors before treatment with 5FU may provide maximal benefit as 5FU-mediated cytotoxicity involves oxidative stress. In line with this notion, Feng and colleagues have devised a nanoliposome-based delivery platform that facilitates delivery of LY294002, a PI3K inhibitor that inhibits autophagy, prior to that of 5FU [56], and reported that nanoliposomes loaded with 5FU and LY294002 augmented cytotoxicity in an ESCC xenograft tumor model as compared to free cocktail combinations. Additional preclinical testing is necessary to determine if this this promising drug delivery strategy may be translated to the clinic to improve response to 5FU, and potentially other chemotherapy drugs, in esophageal cancer patients.

Although the described studies implicate autophagy as a factor that limits apoptosis in response to chemotherapy, autophagy induced by both miR-193b and lithium chloride have been linked to enhanced non-apoptotic cell death in KYSE450 ESCC cells treated with 5FU [51,57]. Additionally, capacity to induce autophagy flux had no relationship to response to the chemotherapeutic agent paclitaxel in a panel of EAC cell lines [58]. In this study, it was further demonstrated that the presence of cytoplasmic p62 alone or in combination with low expression of LC3B was associated with poor response to chemotherapy in EAC patients. As LC3 is induced during autophagy flux while p62 is degraded, the authors attribute therapeutic resistance to functions of p62 beyond its role in autophagy. Indeed, emerging evidence supports various autophagy-independent roles for p62 in carcinogenesis, including induction of the nuclear factor erythroid 2-related factor 2 (NRF2) oxidative stress pathway and activation of mTORC1 [59,60]. It must be noted, however, that immunohistochemical staining for LC3 and p62 may not reflect autophagy flux levels in situ.

Ionizing radiation has also been demonstrated to activate autophagy in ESCC. A protective role of autophagy in response to radiation was demonstrated in vivo as pharmacological autophagy inhibition sensitized Ec9706 and Ec109 ESCC xenograft tumors to irradiation by inducing apoptosis and inhibiting angiogenesis [61,62,63]. Additionally, various ESCC cell lines have been demonstrated to display radiation-induced autophagy that acts to limit apoptosis and cell cycle arrest in vitro [61,64]. Induction of endoplasmic reticulum (ER) stress via tunicamycin has been shown to sensitize EC109 cells to irradiation in vitro, and this was associated with cell cycle arrest, apoptosis and autophagy [65]. ER stress-mediated autophagy in the context of irradiation was induced in response to decreased PI3K/Akt/mTORC1 signaling, and acted in a protective fashion as autophagy inhibition with 3MA augmented cell death in response to radiation combined with tunicamycin. Autophagy has also been demonstrated in the context of eukaryotic elongation factor 2 kinase (eEF2K)-mediated radiation resistance in the ESCC cell line Eca109 [66]; however, it not yet known how autophagy mediates this resistance. Although AMPK/mTORC1/ULK1-mediated activation of autophagy downstream of eEF2K has been demonstrated in breast cancer cells [67], this mechanism has yet to be explored in ESCC cells responding to radiation. High mobility group box 1 (HMGB1) expression is upregulated in ESCC lesions as compared to adjacent normal mucosa and also is associated with tumor recurrence after postoperative radiotherapy (PORT) in ESCC patients [68,69]. HMGB1 has been implicated in radioresistance of several ESCC cell lines via induction of apoptosis and inhibition of autophagy [68,69]. The association between HMGB1 and autophagy in radiation resistance was further supported in ESCC patients as expression of LC3 positively correlated with HMGB1 in tumor tissues and patients with in-field recurrence after PORT exhibited higher LC3 expression compared with those without recurrence [69]. While HMGB1 has been shown to induce autophagy through several mechanisms, including via direct interaction with Beclin-1 [70], how HMGB1 promotes autophagy in ESCC cells responding to ionizing radiation remains to be determined.

Given the refractory nature of esophageal cancer to current standard of care, various experimental therapeutics have been developed. These include molecular targeted therapies, biologics, viral oncolytics, natural compounds, and immunotherapy. In the laboratory setting, several esophageal cancer experimental therapeutics have been associated with alterations in autophagy and/or combined with autophagy modulators.

As overexpression of EGFR has been identified in up to 30% of esophageal tumors [2,3,15,71], therapies targeting the EGFR pathway have been of high interest in esophageal carcinomas. In the ESCC cell lines Ec109 and TE1, the humanized monoclonal anti-EGFR antibody nimotuzumab increased sensitivity to cisplatin and paclitaxel, with induction of autophagy acting to promote cell death [72]. In this context, activation of autophagy with the mTORC1 inhibitor rapamcyin augmented cell death in response to combination of nimotuzumab with either cisplatin or paclitaxel. In EAC, treatment with lapatinib, a small molecule inhibitor of EGFR, was associated with induction of autophagy in OE19 cells in vitro, and pharmacological autophagy inhibition potentiated lapatinib-mediated cytotoxicity [73]. Additionally, basal autophagy flux was elevated in Lapatinib-resistant OE19 cells as compared to parental controls, and cell viability was decreased by autophagy inhibition alone or in combination with lapatinib. These two studies support autophagy modulation as a potential mechanism to improve the efficacy of EGFR-targeted therapies in esophageal cancer; however, careful consideration must be given as activation of autophagy may improve the effects of EGFR-targeted therapies in some contexts and inhibit these effects in other instances. As EGFR-targeted therapies are combined with chemoradiotherapy in clinical trials [74,75,76], future investigations using robust preclinical models should investigate the impact of autophagy modulators on these combination therapies.

Experimental therapeutics targeting facets of esophageal tumor biology beyond EGFR have also been demonstrated to impact autophagy. Several agents targeting members of the Bcl2 family of proteins, which represent key regulators of apoptosis, have been utilized in esophageal cancer therapy with noted effects on autophagy. The synthetic Bcl2 homologous 3 (BH3)-mimetic GX15-070 acts to induce apoptosis by inhibiting activity of anti-apoptotic Bcl2 family members. In the ESCC cell line EC9706, GX15-070 alone induced cell death, and the agent also displayed synergism when combined with 5FU or carboplatin [77]. GX15-070 was also noted to induce autophagy in EC9706 cells, and inhibition of autophagy with 3MA enhanced cell death induced by GX15-70. Activation of autophagy by GX15-070 was associated with an increase in Beclin-1 mRNA, but no change in activation of the PI3K/Akt/mTORC1 signaling axis. Obatoclax, a pan-inhibitor of anti-apoptotic Bcl2 family members, induced cell death associated with AV accumulation in the ESCC cell lines EC109 and HKESC-1 [78]. In this instance, autophagy flux was stalled via obatoclax-mediated downregulation of several cathepsin proteins that act as lysosomal proteases. Given that autophagy and the proteasome are the primary cellular pathways for protein degradation, combining obatoclax with the proteasome inhibitor MG132 induced accumulation of polyubiquinated proteins, which the authors speculated may enhance cancer cell death.

Proton pump inhibitors (PPIs) are the primary therapeutic intervention used in patients with GERD and BE to limit reflux. The PPI Esomeprazole has been demonstrated to induce apoptosis associated with oxidative stress in EAC cell lines, but not the BE cell line CPA [79]. Accumulation of AVs was noted in response to esomeprazole in the EAC cell line OE33 as well as OACM5.1C cells, which were derived from metastatic EAC. Interestingly, esomeprazole induced autophagy flux in OE33 cells, but stalled autophagy in OACM5.1C cells, indicating that the contribution of autophagy to PPI-induced cell death may vary with tumor stage. Dichloroacetate (DCA) is a small molecule inhibitor of pyruvate dehydrogenase kinase that has been shown to exert antitumor effects in preclinical models of breast, colon, and prostate cancer [80,81,82]. In TE1 ESCC cells, DCA treatment induced cell death that was associated with suppressed Akt/mTORC1 signaling and induction of autophagy [83]. Genetic or pharmacological autophagy inhibition augmented cell death in response to DCA alone and in combination with 5FU. Activation of autophagy has also been linked to Endostar, an artificially synthesized anti-angiogenesis drug that improves the efficacy of chemotherapy and chemoradiotherapy in ESCC patients [84,85,86]. In the Eca109 and TE11 ESCC cell lines, Endostar induced autophagy flux via suppression of Akt/mTORC1 signaling. Autophagy activation was found to be protective in response to Endostar as autophagy inhibition via CQ enhanced Endostar-mediated cell death. The mechanism through which Endostar induced cell death has yet to be determined in ESCC, but as this agent acts to inhibit angiogenesis, it would be of interest to determine how Endostar influences ESCC tumor biology in vivo.

Various naturally occurring compounds have also been explored therapeutically in esophageal cancer. Resveratrol is a phytoalexin found in red wine and various foods that has been demonstrated to have anti-tumor effects in human cancer [87]. In the ESCC cell lines EC109, EC9706, and K562, resveratrol induced sub-G1 cell cycle arrest and apoptosis [88]. Activation of autophagy was also noted in response to resveratrol, and genetic or pharmacological autophagy inhibition enhanced resveratrol-mediated cell death. Autophagy in response to resveratrol was characterized by increased expression of Beclin-1 and ATG5, and was independent of AMPK activation. Although the dependence of resveratrol-induced autophagy on Beclin-1 in ESCC cells has not been tested, it is possible that Beclin-1 may be dispensable for this response, as has been demonstrated in breast cancer cells [89]. The ginsenoside Rk3 was recently shown to limit cell proliferation and colony formation in the ESCC cell lines Eca109 and KYSE150 in vitro, as well as KYSE150 tumor growth in vivo [90]. Rk3-mediated cytotoxicity in these cells was associated with G1 cell cycle arrest, apoptosis, and autophagy, the latter attributed to decreased PI3K/Akt/mTOR signaling. 3MA-mediated inhibition of autophagy suppresses cell death in response to Rk3, indicating that autophagy promotes cell death in this context. As plant-based diets have been associated with decreased EAC risk, one study investigated the impact of purified cranberry-derived proanthocyanidin extract (C-PAC) upon EAC cells [91]. C-PAC induced non-apoptotic cell death in the acid-sensitive cell lines JHAD1 and OE33 and the acid-insensitive cell line OE-19; however, autophagy was identified as the mechanism of cell death in acid-sensitive cells, while necrosis killed acid-insensitive cells. In contrast to these in vitro-based findings, OE19 xenograft tumors in mice subjected to oral C-PAC administration displayed decreased tumor growth, as well as evidence of autophagy and suppression of PI3K/Akt/mTORC1 signaling.

In spite of the notion of autophagy as a tumor promoter that is activated in response to stressors, including chemo- and radiotherapy, autophagy may enhance therapeutic efficacy in certain contexts, as demonstrated by several studies described herein. As such, it is critical to understand whether activation or inhibition of autophagy will provide a benefit in esophageal tumors. One key take-away message from this body of work is that context is key when considering how to manipulate autophagy in esophageal cancer. As the process may act to limit or to promote cell death under different conditions (Figure 3), this must be considered as we move toward expanding our knowledge in this area and translating it to the clinical setting.

## 4. Conclusions and Future Directions

The described studies provide compelling evidence for autophagy as a factor contributing to the pathogenesis of esophageal cancer, and support the need for further exploration of autophagy modulators as tools to improve response to both established and experimental therapeutics in esophageal tumors. As autophagy is a dynamic process that is modulated in a context-dependent manner in esophageal cells, future studies defining the role of autophagy in esophageal biology under conditions of health and disease are critical to inform therapeutic strategies targeting autophagy in esophageal cancer.

The use of in vivo models will be a valuable asset in further investigations into autophagy in esophageal carcinogenesis. We have shown that AV levels are augmented in the *L2-Il1b* murine model of BE/EAC [11]. Moreover, Roesly and colleagues have demonstrated that Beclin-1 expression patterns in a rat model of BE/EAC mimic those found in human BE and EAC patients [32]. While in vitro data suggests a protective role for autophagy in esophageal epithelium early in the pathogenesis of the GERD/BE, it would be of great interest to examine how genetic or pharmacological autophagy impacts disease initiation and progression using these in vivo models. Both genetic- and carcinogen-driven models of ESCC have been developed [92,93,94], however, autophagy has yet to be explored in these experimental systems. These models will be of particularly significance when addressing the impact of autophagy modulation on esophageal cancer therapy as responses to autophagy modulators are likely to result from effects on cell types beyond the tumor cells themselves, including fibroblasts and immune cells.

Overall, the current literature indicates that autophagy may promote esophageal cell survival or cell death in a context-dependent manner. Adding a layer of complexity, enhanced autophagy flux or stalled autophagy may contribute to either autophagy-mediated survival or cell death. When thinking about how to best utilize autophagy inhibitors for cancer therapy, it will be critical to accurately define (1) whether autophagy is activated or stalled in human patients; and (2) how modulation of autophagy will impact cancer cells. Utilizing a flow cytometry-based assay for AV detection in human and mouse tissues incubated ex vivo in the presence or absence of the autophagy flux inhibitor chloroquine (CQ) [12,95], we have developed a novel autophagy flux assay that may be used to accurately determine autophagy flux status and its relationship to esophageal carcinogenesis. Additionally, 3D ESCC organoid culture [49] may serve as an ex vivo platform to test the impact of autophagy inhibitors upon cancer cells in the context of precision medicine. Furthermore, the development of agents with the ability to specifically and potently activate or inhibit autophagy is of great interest in order to interrogate the role of autophagy in cancer biology and also to limit non-specific effects in human patients.

## Figures and Tables

**Figure 1 cancers-11-01697-f001:**
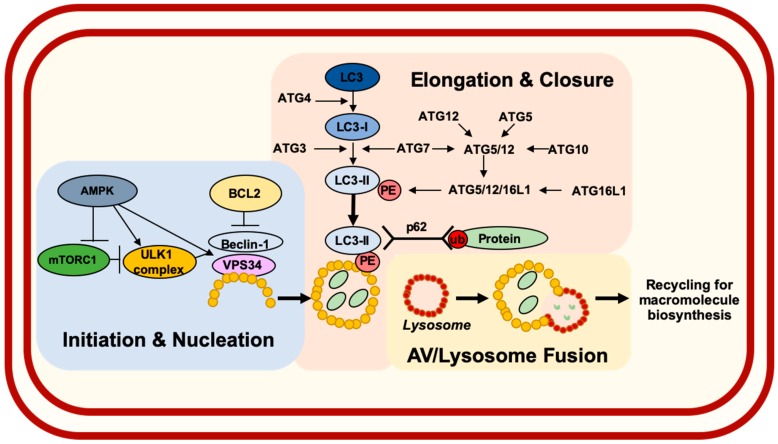
Molecular regulation of autophagy. Mammalian target of rapamycin 1 (mTORC1) acts a critical negative regulator of autophagy under nutrient-rich conditions. AMP-activated kinase (AMPK) serves a key positive regulator of autophagy in response to energy depletion. AMPK promotes AV initiation and nucleation through assembly of the Unc51-like kinase 1 (ULK1) complex. For nucleation to continue, Beclin-1 must dissociate from Bcl2 in order to interact with vacuolar sorting protein (VPS)34, a class III PI3 Kinase. AV elongation to surround p62/SQSTM1-associated cargo proteins involves cleavage of Microtubule-associated protein light chain 3 (LC3) by autophagy-related (ATG)4, generating LC3-I. LC3-I is then lipidated (generating LC3-II) through addition of phosphatidylethanolamine (PE) by two ubiquitin-like conjugation systems consisting of various ATGs. Following closure, AVs undergo fusion with lysosomes where acid hydrolase enzymes break down autophagic cargo so that their constituents can be used for biosynthesis of macromolecules.

**Figure 2 cancers-11-01697-f002:**
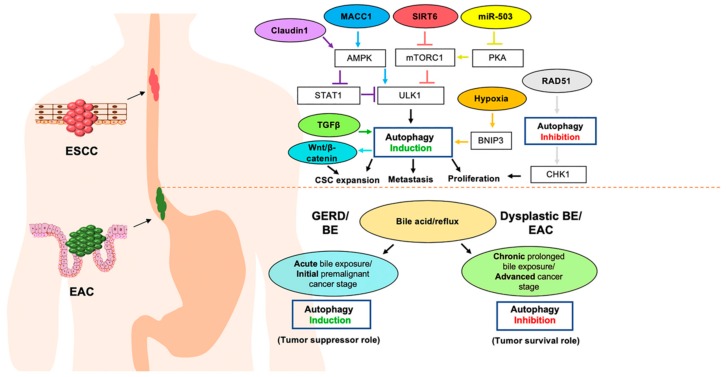
Alterations in autophagy in response esophageal cancer-associated factors. Esophageal squamous cell carcinoma (ESCC) occurs in the proximal portion of the esophagus as esophageal keratinocytes undergo malignant transformation. The schematic above the dotted line depicts how the listed cellular and environmental factors have been demonstrated to impact autophagy in ESCC as well as the effects of autophagy activation and inhibition upon ESCC cells. Esophageal adenocarcinoma (EAC) occurs in the distal portion of the esophagus as esophageal epithelium is displaced by a specialized intestinal metaplasia, Barrett’s esophagus (BE), in response to gastroesophageal reflux disease (GERD). The schematic below the dotted line depicts how bile/acid reflux impacts autophagy in the early premalignant states of GERD and BE as well as in dysplastic BE and EAC.

**Figure 3 cancers-11-01697-f003:**
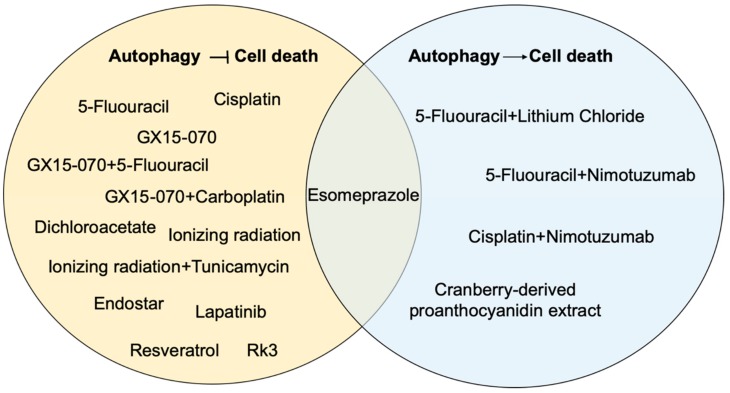
Summary of studies investigating effects of autophagy on therapeutic response in esophageal cancer. Therapeutic interventions that activate autophagy to limit cell death are shown in the yellow circle, while therapies that activate autophagy to promote cell death are shown in the blue circle. Esomeprazole is present in the overlapping space between the two circles as it has been shown to activate autophagy to limit cell death and also promote cell death in a context-dependent manner.

**Table 1 cancers-11-01697-t001:** Autophagy markers showing significant association with survival in esophageal cancer

Risk Factor	Expression Description	*n*	Prognostic Relevance	References
ESCC				
LC3 *	High/Postive	150	(M, OS) OR 1.657, 95% CI; 1.048–2.622	[35]
		43	Decreased overall survival (*p* = 0.032)	[36]
		129	Decreased overall survival (*p* = 0.0382)	[26]
		142	Increased overall survival (*p* = 0.04)	[38]
	Negative	118	Decreased overall survial (*p* = 0.021)	[43]
Beclin-1	Negative	54	Decreased overall survival (*p* = 0.004)	[44,45]
		118	(M, OS) HR 0.511, 95% CI; 0.299–0.874	[43]
ULK1	High	248	(M, OS) RR 2.220, 95% CI: 1.434–3.436	[46]
	Low	86	(U, OS) HR 1.754, 95% CI: 1.022–3.010	[45]
LC3, Beclin-1	Positive, Positive	150	Decreased overall survival (*p* = 0.038)	[35]
LC3A, p53	High, High	114	(M, OS) HR 2.8, 95% CI: 1.536–6.183	[39]
EAC				
LC3	Low (Diffuse Cytoplasmic)	104	Decreased overall survival (*p* < 0.001)	[41]
	High (Crescent or Ring-like)	104	Decreased overall survival (*p* = 0.02)	
	High (Globular)	104	(M, OS NaN) HR 6.086, 95% CI: 3.179–11.653	
p62 (Cyto/Nuc)	Low/Low	116	(M, OS) HR 0.561, 95% CI: 0.329–0.956	[42]
LC3B, p62	Low, Low	116	(M, OS) HR 0.549, 95% CI: 0.330–0.914	

* Note: LC3 is assessed using antibodies against LC3, LC3A, LC3B, or cleaved LC3A; CI—confidence interval; Cyto—cytoplasmic; EAC—esophageal adenocarcinoma; ESCC—esophageal squamous cell carcinoma; HR—hazard ratio; LC3—microtubule associated protein light chain 3; M—multivariate analysis; NaN—Neoadjuvant Naïve; NaT—neoadjuvant therapy; Nuc—nuclear; OR—odds ratio; OS—overall survival; RR—relative risk; U—univariate analysis; ULK1—Unc-51-like autophagy activating kinase 1.

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
