# Peer review of "Roles for Autophagy in Esophageal Carcinogenesis: Implications for Improving Patient Outcomes"

_cancers, 2019, doi:10.3390/cancers11111697_

Round 1
Reviewer 1 Report
In this manuscript by Saxena et al., the authors focused on the relationship between autophagy and esophageal cancer. They also
discussed considerations for the design of autophagy-targeting therapeutic strategies with potential to improve outcomes for esophageal
cancer patients. The latest research progress was reviewed. This manuscript provided interesting information and may help future clinical
application.
Author Response
In this manuscript by Saxena et al., the authors focused on the relationship between autophagy and esophageal cancer. They also
discussed considerations for the design of autophagy-targeting therapeutic strategies with potential to improve outcomes for esophageal
cancer patients. The latest research progress was reviewed. This manuscript provided interesting information and may help future clinical
application.
We thank the Reviewer for careful consideration of the manuscript and the kind words.
Reviewer 2 Report
The present review is quite comprehensive of the state of the art studies on the subject and could stimulate the discussion as well as the development of more precise scoring protocols to evaluate the impact of autophagy activation/suppression in this type of cancer. I just suggest to revise the punctuation along the text, as some sentences are very long, devoid of commas and difficult to read.
Author Response
The present review is quite comprehensive of the state of the art studies on the subject and could stimulate the discussion as well as the development of more precise scoring protocols to evaluate the impact of autophagy activation/suppression in this type of cancer. I just suggest to revise the punctuation along the text, as some sentences are very long, devoid of commas and difficult to read.
We thank the Reviewer for careful consideration of the manuscript and have carefully edited the text (changes are marked in Red) to make it easier to read.
Reviewer 3 Report
In the article, the authors described the significance of autophagy in esophageal carcinogenesis and its influence on improving patient outcomes. The esophageal cancer is the most aggressive type and a 5-year survival rate for this particular cancer is very low. Therefore a full characteristics of carcinogenesis and cancer treatment especially of the new autophagy related mechanism of those is necessary. The topic of the publication is extremely absorbing. In particular, the role of autophagy in cancer treatment has not been fully characterized in the literature as of yet and in many cases is disputable.
However, I have a few minor revisions:
Figure 1 is not described in the text of the manuscript. It should be done briefly but accurately. In figure 1, the stages of autophagy should be marked e.g. by using other colour of background. The titles of section 2.2 and 2.3 should be changed. They are too general. Additionally, section “Insights from studies in human tissues” does include more than only studies conducted on human tissues. The authors divided the 3 sections into two parts: Autophagy in chemo- and radiotherapy and Autophagy in experimental therapeutics for esophageal cancer. For better characterisation of authophagy problem in oncotherapy one needs to distinguish two aspects of response to therapy. The type of treatment is of lesser importance at this point. Discussion section should be slightly rearranged. The authors should concentrate on the roles for autophagy in esophageal carcinogenesis and patient treatments. The successful formation of 3D organoids from ESCC patients constitutes a vital achievement, however it is not the major subject matter of the article.Author Response
In the article, the authors described the significance of autophagy in esophageal carcinogenesis and its influence on improving patient outcomes. The esophageal cancer is the most aggressive type and a 5-year survival rate for this particular cancer is very low. Therefore a full characteristics of carcinogenesis and cancer treatment especially of the new autophagy related mechanism of those is necessary. The topic of the publication is extremely absorbing. In particular, the role of autophagy in cancer treatment has not been fully characterized in the literature as of yet and in many cases is disputable.
We thank the Reviewer for careful consideration of the manuscript and appreciate the kinds words. We have addressed the noted minor revisions as detailed below and provide an updated version of the manuscript with all changes marked in red.
1-Figure 1 is not described in the text of the manuscript. It should be done briefly but accurately.
We now explicitly describe Figure 1 in the text.
2-In figure 1, the stages of autophagy should be marked e.g. by using other colour of background.
We have edited Figure 1 to include color-associated markings of the various stages of autophagy.
3-The titles of section 2.2 and 2.3 should be changed. They are too general.
We have removed all subheadings from each section.
4-Additionally, section “Insights from studies in human tissues” does include more than only studies conducted on human tissues.
We have removed all subheadings from each section.
5-The authors divided the 3 sections into two parts: Autophagy in chemo- and radiotherapy and Autophagy in experimental therapeutics for esophageal cancer. For better characterisation of authophagy problem in oncotherapy one needs to distinguish two aspects of response to therapy. The type of treatment is of lesser importance at this point.
We have removed all subheadings from each section.
6-Discussion section should be slightly rearranged. The authors should concentrate on the roles for autophagy in esophageal carcinogenesis and patient treatments. The successful formation of 3D organoids from ESCC patients constitutes a vital achievement, however it is not the major subject matter of the article.
We have re-arranged the discussion to first convey the overall conclusions gleaned from the literature review, followed by a discussion of future directions to improve our understanding of autophagy in esophageal biology and carcinogenesis.